# Development and Stability of a New Formulation of Pentobarbital Suppositories for Paediatric Procedural Sedation

**DOI:** 10.3390/pharmaceutics15030755

**Published:** 2023-02-24

**Authors:** Matthieu Lebrat, Yassine Bouattour, Coralie Gaudet, Mouloud Yessaad, Mireille Jouannet, Mathieu Wasiak, Imen Dhifallah, Eric Beyssac, Ghislain Garrait, Philip Chennell, Valérie Sautou

**Affiliations:** 1CHU Clermont-Ferrand, Pôle Pharmacie, F-63003 Clermont-Ferrand, France; 2Université Clermont Auvergne, CHU Clermont Ferrand, Clermont Auvergne INP, CNRS, ICCF, F-63000 Clermont-Ferrand, France; 3Université Clermont-Auvergne, UFR Pharmacie, UMR MEDIS, F-63001 Clermont-Ferrand, France

**Keywords:** pentobarbital, formulation, suppositories, stability, sedation

## Abstract

Pentobarbital is a drug of choice to limit motion in children during paediatric procedural sedations (PPSs). However, despite the rectal route being preferred for infants and children, no pentobarbital suppositories are marketed, and therefore they must be prepared by compounding pharmacies. In this study, two suppository formulations of 30, 40, 50, and 60 mg of pentobarbital sodium were developed using hard-fat Witepsol^®^ W25 either alone (formulation F1) or with oleic acid (formulation F2). The two formulations were subjected to the following tests described in the European Pharmacopoeia: uniformity of dosage units, softening time, resistance to rupture, and disintegration time. The stability of both formulations was also investigated for 41 weeks of storage at 5 ± 3 °C using a stability-indicating liquid chromatography method to quantify pentobarbital sodium and research breakdown product (BP). Although both formulae were compliant to uniformity of dosage, the results were in favour of a faster disintegration of F2 compared to F1 (−63%). On the other hand, F1 was found to be stable after 41 weeks of storage unlike F2 for which several new peaks were detected during the chromatographic analysis, suggesting a shorter stability of only 28 weeks. Both formulae still need to be clinically investigated to confirm their safety and efficiency for PPS.

## 1. Introduction

Paediatric Procedural Sedation (PPS) is often required for children who undergo a radiologic imaging exam. Indeed, the patients need to stay motionless during the image acquisition sequence, which can be difficult for paediatric patients. Two pharmacological options are available to obtain the patient’s stationary state: a general anaesthesia or a sedation. The latter option does not require resources, which can be limited, such as the presence of an anaesthesiologist or an operating theatre [1,2,3,4]. Nonetheless, almost all the nondissociative drugs that can be used as sedatives in PPS (such as opioids, benzodiazepines, barbiturates (including pentobarbital), etomidate, and propofol) can lead to general anaesthesia with the loss of airway reflexes, depending on the dosage [2]. Their judicious use by trained health-care providers must therefore be well monitored. In the last century, there was rapid growth of PPS with practitioners increasing in skills, an intensification of outpatient settings, and the widespread ethical imperative of managing patient anxiety [5].

The core purpose of PPS for image acquisition in an outpatient setting is to limit invasive acts; thus, an orally or rectally administered drug is needed. Oral chloral hydrate has been an option for many years, but it induces effects that are unreliable for children older than 3 years old [2] and is a potentially carcinogenetic drug that justifies restricting its use according, for example, to the French drug regulatory agency (ANSM) [6], even if other agencies, such as the EMA or FDA, have not as yet published any reservations concerning its safety. In addition, according to a study by Mason et al. (*n* = 1316 patients), it has a higher incidence of adverse events than pentobarbital in infants during imaging [7].

The rectal (transmucosal) route is particularly convenient for PPS, especially for infants, because it is not invasive, does not pose any choking risks, and does not need to be taste-masked [8]. It also has the advantage of a rapid and more predictable onset than the oral route [9]. Rectally administered drugs, which are described in the literature as being possible to use for PPS, are benzodiazepines, chloral hydrate, ketamine, and barbiturates; however, according to Lam et al. [9] in their 2018 systemic review, rectal midazolam (the main benzodiazepine used for PPS) was the least effective of these drugs, and ketamine induced very common adverse effects (hypersalivations and hallucinations, for example). They reported a sedating efficacy of 60–75% for midazolam in most trials versus 80–95% for the other drugs. The only other rectal option appears to be pentobarbital, which was the third most used sedative for head computerized tomography in children in 2017 in the US (cohort of *n* = 24,418 patients [10]. Apart from the US, pentobarbital in the rectal form is also used in France, but it has not been used in the UK since its withdrawal in the 1960s due to its potential for abuse [11] and therefore is not mentioned in the NICE Sedation in children and young people guideline, updated in 2019 [12]. However, as part of their systematic review of needle-free pharmacological sedation techniques in paediatric patients, de Rover et al. reaffirmed the interest and need for better alternative sedation techniques to the I.V. or I.M. route in the imaging procedure context [13].

Pentobarbital is a barbiturate acting as a positive allosteric modulator of γ-aminobutyric acid (GABA) receptors when it interacts with its binding site, which is distinct from the GABA and benzodiazepine ones. Pentobarbital potentiates the effect of GABA on its receptor and increases the mean open duration of the GABA-A receptor channel. Unlike benzodiazepines, at high doses, it directly activates the chloride channel. It has also been reported that barbiturates block the α-amino-3-hydroxy-5-methylisoxazol-4-propionate (AMPA) and kainite receptors, which are glutamate receptors [14]. Its paediatric dosing for the oral or rectal route is usually of 3–6 mg/kg, maximum 100 mg per dose (for patients younger than 4 years old) and 1.5–3 mg/kg, maximum 100 mg per dose (if older than 4 years old), but it should be avoided in patients with porphyria and may produce paradoxical excitement, which is a common adverse event (1–10% frequency) [9,15].

Because pentobarbital is not marketed in any dosage form allowing a rectal administration, formulations of pentobarbital (and more specifically of pentobarbital sodium, as the salt form is readily available in pharmaceutical grade), such as those made by hospital pharmacists, are needed. Belo et al. [16] recently developed a hydrogel formulation for rectal administration containing pentobarbital sodium at a concentration of 25 mg/mL. However, this formulation presented some disadvantages, such as containing sodium benzoate, which is an excipient with a known effect, especially on hormone levels and on inflammation, and can also have an irritating action on the gastric mucosa [17]. Furthermore, hydrogels need to be stored as stock solutions and then divided into single doses to be given to patients. In addition, they also need a special device to be administered, which could complexify the administration process. On the other hand, suppositories are a well-known and greatly used dosage form, especially for the paediatric population, notably because of their manufacturability and established quality control procedures as well as their ease of administration and possible systemic effects [8]. The use of pentobarbital sodium suppositories has already been described (for the sedation of children undergoing auditory brainstem response testing) [3], but the details of the formulation and its stability are not publicly available.

The objective of this study was therefore to present and compare the pharmaceutical formulation and stability properties of two different formulations of pentobarbital sodium suppositories for paediatric procedural sedation.

## 2. Materials and Methods

### 2.1. Chemicals

Pharmaceutical pentobarbital sodium powder (CAS 57-33-0, batch 2019100735, exp. 30/12/2023) and oleic acid (CAS 112-80-1, batch 20002698002, exp. 31/10/2022) were supplied by Inresa (Bartenheim, France). Witepsol W25 (batch 19090001/A, exp. 05/2023) was supplied by Cooper (Melun, France). Deionised water was purchased from Fresenius (Sèvres, France). The High-Performance Liquid Chromatography (HPLC)-grade methanol was obtained from Carlo Erba reagents (CAS 67-56-1, Val de Reuil, France). Hydrogen peroxide 30% was supplied by Cooper (CAS 7722-84-1, Melun, France). Potassium dihydrogen phosphate (CAS 7778-77-0; KH_2_PO_4_) was purchased from Sigma-Aldrich (St. Louis, MO, USA). HCl (CAS 7647-01-0), NaOH (CAS 1310-73-2), and 1-Octanol (CAS 111-87-5) were purchased from Honeywell via a local vendor (MC2, Clermont-Ferrand, France).

### 2.2. Formula Determination and Preparation

All compounds used were of pharmaceutical grade, and the pentobarbital that was weighed, used, and quantified was in its sodium salt form. Two formulations were developed to prepare suppositories with a total mass of 1.1 g, containing 30, 40, 50, and 60 mg of pentobarbital sodium and Witepsol^®^ W25 as a base either alone (formulation F1) or with oleic acid added to solubilize pentobarbital in Witepsol^®^ W25 (formulation F2). The range of strengths (from 30 mg to 60 mg of active ingredient) was chosen according to pentobarbital paediatric dosing (3–6 mg/kg, if <4 years old, max. 100 mg; 1.5–3 mg/kg, if >4 years old, max. 100 mg) to cover different paediatric weights starting approximately at 1 year old.

The suppositories were prepared using 1 g single-use plastic moulds, which could contain 1.1 g of Witepsol^®^ W25 when completely filled. The amounts of oleic acid used in F2 were determined by calculating the displacement factor (f) in the Witepsol^®^ W25 base [18] (see Appendix A). For F1, because the formulation contained pentobarbital sodium at a concentration equal to or lower than 5% of the total weight of the suppository, f was considered negligible [19]. Table 1 gives the details of the quantities used for each formulation (per suppository). To obtain the exact quantity of the pentobarbital base, the mass of pentobarbital sodium must be multiplied by 0.911 (as the molar masses of pentobarbital sodium and pentobarbital are 248.25 g/mol and 226.27 g/mol, respectively).

The components were weighed to prepare batches of 70 units. For F1, pentobarbital sodium powder was triturated with about half the needed quantity of Witepsol^®^ W25 in a mortar, then transferred into a glass beaker. This mixture was melted for 10 min in a double boiler at 50 °C. In another double boiler, the rest of the Witepsol^®^ W25 was also melted at 50 °C for 10 min; then, it was added to the beaker containing the pentobarbital sodium. The preparation was mixed for 20 min using an agitation rod, with the temperature maintained at 50 °C. For F2, the suppositories were prepared by melting the Witepsol^®^ W25 in a double boiler at 80 °C; then, the pentobarbital sodium powder was slowly added under gentle agitation until the mixture reached a milky state. The oleic acid was then added drop by drop, whilst still mixing the preparation until obtaining a clear yellow solution.

Finally, the suppositories were cast in single-use plastic moulds (LGA S.A.S, La-Seyne-Sur-Mer, France) whilst still being stirred regularly. The suppositories were left 45 min at room temperature before being stored in a refrigerator (5 ± 3 °C) for a minimum of 24 h before the analyses were performed.

### 2.3. Characterisation of the Suppositories

After the preparation by experienced operators of three batches (each batch containing 70 units) per dosage and per formulation, the suppositories were subjected to the below-mentioned pharmaceutical technical procedures recommended by monograph 1145 Rectal Preparations of the European Pharmacopoeia (Ph. Eur.), with *n* = 3 (one from each batch) unless specified otherwise.

#### 2.3.1. Uniformity of Dosage Units

The analysis of the uniformity of the units in both formulae was performed according to monograph 2.9.40 “Uniformity of dosage units” of the Ph. Eur. on each prepared batch [20]. After sampling 10 units per dosage form and pentobarbital sodium extraction (see Section 2.4.2.2), the amount of pentobarbital sodium in each suppository was determined by measuring the absorbance at 240 nm by UV-visible spectrophotometry using a Jasco V-670 spectrometer (Jasco Corporation, Lisses, France) with a quartz measuring cell. The calibration was validated following the preparation of 3 calibration curves and 18 control points each day for three days. The linearity was obtained for pentobarbital sodium concentrations of 12 to 28 µg/mL. The matrix effect (investigated by comparing calibration curves obtained with and without excipients) was found to be negligible. The equation of the regression curve that was used was Y = 0.0350471X + 0.0443862 with Y being the absorbance measured at 240 nm and X the pentobarbital sodium concentration in the aqueous phase after extraction and dilution in deionized water (µg/mL). The determination coefficient R^2^ was 0.998. The experimental acceptance values (EAVs) of the 10 quantified units were calculated and should not be higher than 15%. Suppositories of compliant batches were selected for further testing.

In addition, to compare the ease of preparation between F1 and F2, four operators with no previous experience whatsoever in suppository preparation produced (following only the preparation instructions sheet) one batch of the 30 mg dosage of each formula following this order: two operators prepared F1 then F2 batches, and conversely, the other two operators started with F2 then F1, in order to reduce the learning bias. The EAVs for each batch were calculated, and compliance with 2.9.40 monography was assessed.

#### 2.3.2. Resistance to Rupture/Hardness

The hardness measurements were performed using an SBT apparatus (Erweka, Germany) at room temperature (22 ± 0.5 °C) to evaluate the mechanical resistance of the suppositories to crushing, i.e., to determine the mass at which the suppository breaks or crushes at room temperature [21]. One suppository of each dosage per formula was placed between two jaws applying a mass of 600 g, and a mass of 200 g was added to the rod attached to the upper jaw every minute until the suppository was crushed. The time taken to crush each tested suppository was recorded. This test was adapted from monograph 2.9.24 of the Ph. Eur. [22].

#### 2.3.3. Softening Time Determination of Lipophilic Suppositories

The softening time was measured following monograph 2.9.22 of the Ph. Eur. (apparatus A) [23]. Briefly, each tested suppository was introduced by its tip into a glass tube containing 10 mL of water and placed in a water bath equilibrated at 36.5 ± 0.5 °C. A rod was immediately introduced after the suppository introduction. After the cover was put on the tube (this being the start of the time measurements), the time that elapsed until the rod sank down to the bottom of the glass tube and the mark ring reached the upper level of the plastic cover was noted.

#### 2.3.4. Disintegration Test

The disintegration tests were performed following recommendations adapted from monograph 2.9.2 of the Ph. Eur., using a reciprocating cylinder dissolution apparatus (BIO-DIS reciprocating cylinder apparatus, USP Apparatus 3, Agilent, USA), at 37 °C using 250 mL phosphate buffer solution at pH 6.8 as dissolution medium, at 15 dips per minute during 15 min. The disintegration time was determined visually and confirmed by measuring the concentration of pentobarbital sodium released in the phosphate buffer solution at that time, using the UV-visible spectrophotometric method described in Section 2.3.1.

### 2.4. Stability Study: Design and Analyses

The stability of the pentobarbital sodium suppositories was studied for 41 weeks on one batch of 30 mg and one batch of 60 mg suppositories of each formulation conditioned in unopened, single-use plastic moulds. The pentobarbital suppositories were stored in a refrigerator (Liebherr, Bulle, Switzerland) at 5 ± 3 °C monitored every day until analysis.

After preparation (day 1) and after 2, 4, 16, 28, and 41 weeks, five units (ten for day 1) of each formulation and dosage form were submitted to a visual inspection, pentobarbital quantification and breakdown products (BPs) research (i.e., looking specifically for products resulting from the degradation of pentobarbital).

#### 2.4.1. Visual Inspection

The suppositories were opened from their single-use plastic mould into glass test tubes and were visually inspected under daylight and under polarized white light from an inspection station (LV28, Allen and Co., Liverpool, UK). The appearance and colour of the suppositories were noted, and a screening for visible inhomogeneity or colour change was performed.

#### 2.4.2. Pentobarbital Quantification and Breakdown Product Research

##### 2.4.2.1. Preparation of Pentobarbital Standard Solutions

Pentobarbital stock solutions (150 µg/mL) were prepared by accurately weighing 3 mg of pentobarbital sodium. The volume was made up to the mark with deionized water in 20 mL volumetric flask. The solutions were stored in the dark at 5 ± 3 °C for a maximum of 7 days (stability for 7 days confirmed in aqueous solution by Anderson et al. [24]).

##### 2.4.2.2. Pentobarbital Extraction Method

The data concerning the validation of the extraction method are provided in the Appendix A. Each suppository was put into an Erlenmeyer flask and melted using a water bath at 50 °C. Then, 10 mL of 1-Octanol was added to the Erlenmeyer flask and ultrasonicated for 15 min. A total of 20 mL of 0.1 N NaOH solution was then added, and the mixture was homogenized before the Erlenmeyer flasks were then emptied into a separating funnel, previously rinsed with the 0.1 N NaOH solution. The separating funnels were inverted 20 times and left to decant at room temperature. After 24 h, the lower aqueous phase was withdrawn for UV-visible spectrophotometry analysis, after performing appropriate dilution in deionized water (depending on the suppository dosage to be in the range of linearity (see previous Section 2.3.1)). For HPLC analysis, the lower phase was diluted 1/10th to obtain the target concentration of 150 µg/mL.

##### 2.4.2.3. Stability Indicating UV-DAD Chromatographic Method

The chromatographic analyses were performed on a Prominence-I LC-2030C 3D (Shimadzu France SAS, Marne La Vallée, France) using LabSolutions^®^ Lite software (5.82 version, Shimadzu France SAS, Marne La Vallée, France) to collect and process the data (e.g., peak time and peak area). The compact system contained an integrated degasser, an analytical pump, a thermostated autosampler, and a thermostated column compartment. Diode array detection from 190 nm to 800 nm was used to assess peak purity and detect breakdown products. Pentobarbital quantification was performed at 214 nm. The chromatographic separation was performed on an EC 250/4.6 Nucleodur C18 HTec column (250 × 4.6 mm, 5 µm particle size; Macherey-Nagel, Düren, Germany). The pentobarbital content in the suppositories was determined using a stability-indicating HPLC method slightly adapted from a previously published method by Ajemni et al. [25]. In brief, the stationary phase (C18) and the mobile phase (0.01 M phosphate buffer pH3 adjusted with HCl (Phase A) and methanol (Phase B), 40:60, *v*/*v*) were used, but we switched the isocratic elution to a gradient one (Table 2), at a flow rate of 1.2 mL/min; injection volume was 25 µL. The column oven and rack temperature were set at 40 °C to liquefy the base of our suppositories in case some particles were still present after the liquid–liquid extraction.

The matrix effect was evaluated by comparing three calibration curves (slopes and intercepts) obtained from sodium pentobarbital only (pharmaceutical quality) with those containing sodium pentobarbital in the presence of all the excipients.

Linearity was initially confirmed by preparing one calibration curve daily for three days using five concentrations of sodium pentobarbital at 10, 100, 200, 300, and 350 μg/mL, diluted in deionized water. The calibration regression curves were weighted by 1/C (C = concentration of pentobarbital) to minimize the weight of extreme points on the prediction of the mathematical model. Each calibration curve should have a determination coefficient R^2^ equal to or higher than 0.999. Homogeneity of the curves was verified using a Cochran test. ANOVA tests were applied to determine applicability. Each day for three days, six solutions of pentobarbital sodium 150 µg/mL were prepared, analysed, and quantified using a calibration curve. To verify the method’s precision, repeatability was estimated by calculating the mean relative standard deviation (RSD) of intraday analysis, and intermediate precision was evaluated using an RSD of interday analysis. Both RSDs should be less than 5%. Specificity was assessed by comparing the UV spectra DAD detector of each analysis to a pure sodium pentobarbital UV spectrum (raw material for pharmaceutical use). Method accuracy was demonstrated by evaluating the recovery of five theoretical concentrations to experimental values found using the mean curve equation, and results should be found within the range of 95–105%. The overall accuracy profile was constructed according to Hubert et al. [26,27,28].

In order to exclude potential interference of degradation products with pentobarbital quantification, sodium pentobarbital 150 µg/mL stock solutions were subjected to the following forced degradation conditions: 10.8 N hydrochloric acid for 2, 19, 24, 72, and 192 h at 25 °C and 50 °C; 10.8 N sodium hydroxide for 2, 24, 48, 72, and 192 h at 25 °C and 50 °C; 30% hydrogen peroxide for 7 days at 50 °C; 3% hydrogen peroxide for 7 days at 50 °C and thermal degradation for 7 days at 80 °C. Susceptibility to light was analysed 3 times after solution preparation for up to 7 days using an UVA light (5.83 Mlux/h) in climatic chamber at 25 °C (Binder GmbH, Tuttlingen, Germany).

#### 2.4.3. Data Analysis—Acceptability Criteria

The stability of pentobarbital sodium suppositories was assessed using the following parameters: visual aspect of the suppositories, pentobarbital concentration, and presence or absence of BPs.

The study was conducted following the methodological guidelines adapted from the International Conference on Harmonisation for stability studies [29], notably topic Q1A (R2). A variation of concentration outside the 90–110% range of the initial pentobarbital quantity (including the limits of a 95% confidence interval of the measures) could be considered as a sign of instability after verification of the dosage content in all units. The presence of BPs and the variation of the physicochemical parameters were also considered a sign of pentobarbital instability. The observed pentobarbital suppositories must be homogenous and of unchanged colour.

## 3. Results

### 3.1. Characterisation of the Suppositories

All results of the characterization of both formulations are presented in Table 3. We found that all batches were compliant with monograph 2.9.40 (EAV < 15%) when prepared by experienced operators. When oleic acid was used (formulation F2), we noticed a reduction (−63% on average) in the softening time for all dosages, ranging, for example, from 8 min 15 s for formulation F1 to 3 min 20 s for formulation F2 (for the 30 mg dosage). We found that the resistance to rupture of the suppository and the disintegration time were also reduced in F2 (−80% and −42%, respectively, on average for all dosages).

On the other hand, the preparation of suppositories by operators with no experience demonstrated that the success rate for the production of suppositories with F2 was 100% versus 25% for the F1. The production process of suppositories with oleic acid was also found to be simpler. The uniformity of dosage results are presented in Table 4.

### 3.2. Pentobarbital Quantification and Breakdown Product (BP) Research: Method Validation Results

The pentobarbital retention time was 7.2 ± 0.2 min (Figure 1). The chromatographic method used was found to be linear for concentrations ranging from 10 to 350 μg/mL. The average weighted regression equation was Y = 34,854X + 22,873, where X is the pentobarbital concentration in the aqueous phase after extraction (in μg/mL), and Y is the surface area of the corresponding chromatogram peak. The average determination coefficient R^2^ of the three calibration curves was 0.999. No matrix effect was detected, as the sodium pentobarbital retention time, the slope, and the intercept of the calibration curve did not change significantly with the matrix.

The relative mean trueness bias coefficients were less than 1.5%, except for the 100 μg/mL calibration point, for which it was 1.86%. The mean repeatability RSD coefficient and mean intermediate precision RSD coefficient were less than 2.91%. The accuracy profile constructed with the data showed that the limits of the 95% confidence interval coefficients were all within 5% of the expected value, except for the 300 and 350 μg/mL calibration points, for which the lower range limits were −5.23% and −5.52%, respectively. The detection limit was theoretically calculated to be 0.5 μg/mL, yet the experimental signal-to-noise ratio at that concentration was 90, thus indicating that a lower detection limit could be reached. The limit of quantification was theoretically calculated and then fixed at 1.5 μg/mL (experimental signal-to-noise ratio of 218) with a mean relative bias of −9.47%.

The reference chromatogram of 150 µg/mL sodium pentobarbital at 214 nm is well defined with no visible impurities as shown in Figure 1.

After forced degradation (see the detailed results presented in Table 5) in highly stressful conditions, BPs were detected, particularly in alkaline-forced conditions where after 48 h and 7 days at 10.8 N of NaOH at 50 °C, 9.7% and 30.9% of pentobarbital degradation, respectively, was found. No degradation was detected after 8 days of 10.8 N of HCl at 50 °C. No BPs were detected when pentobarbital solutions were exposed to UV-vis light for 7 days at 25 °C or placed in oxidative conditions with 30% H_2_O_2_ at 50 °C for 7 days. After 7 days at 80 °C, a loss of 11.2% of pentobarbital was noticed, with four breakdown products detected at 3.3, 11.0, 11.8, and 12.3 min. All BPs were detected with a resolution higher than 1.5 from the pentobarbital peak.

The chromatograms obtained from these forced degradations are shown in Figure 2.

### 3.3. Physical Stability

All samples stayed homogeneous and white. Their appearance under daylight and polarized light was unchanged during the study for both tested formulations, and there was no visible change of colour or inhomogeneity.

### 3.4. Chemical Stability

The statistical pentobarbital quantity distribution of the 10 initial samples of each batch is illustrated below with their box plot in Figure 3. The increased variability of the formulation F1 (without oleic acid) was again confirmed.

Throughout the stability study, for the 30 mg pentobarbital suppositories with and without oleic acid conditions, the mean pentobarbital quantities did not vary by more than 7.01% (95%CI: 105.97–108.05% of the mean initial quantity, at the 16-week time point), as presented in Figure 4A. For the 60 mg pentobarbital suppositories without oleic acid, the mean pentobarbital quantities did vary by 14.96% from the mean initial quantity (95% CI: 112.89–117.03%) at the 28-week time point, as presented in Figure 4B. For the 60 mg pentobarbital suppositories with oleic acid, the mean pentobarbital quantities did not vary by more than 6.97% (95%CI: 104.88–109.06 of the mean initial quantity), except after 2 weeks of storage for which an isolated decrease in the amount of pentobarbital in the tested suppositories was detected, without the emergence of any BP.

The chromatograms showed no sign of previously identified pentobarbital BPs during the length of study for both types of pentobarbital formulation at 5 ± 3 °C (some representative chromatograms are presented in Figure 5). Nonetheless, we can see that on the chromatograms of suppositories with oleic acid, small peaks were detectable at the 41-week time point (Figure 5A,B with retention times between 2.5 and 5.5 min). The peaks do not have retention times corresponding to the detected BPs produced during forced degradation. Throughout the study, the purity of the pentobarbital peak was confirmed by spectral analysis and was 100% (see Appendix A for an example after 41 weeks of storage).

## 4. Discussion

In this work, we studied two formulations of pentobarbital suppositories for potential use in paediatric patients for PPS. The results show that whilst formulation F2 containing oleic acid possessed characteristics that could favour its use in PPS, such as faster softening and disintegration times compared to formulation F1, as well as being easier to produce, the materialisation of unknown compounds over time, especially at the 41-week time point, does raise questions about its stability past 28 weeks, which was not the case for formulation F1.

Because oral pentobarbital could have poor palatability with a high refusal or vomiting risk among children, the rectal route (which has been demonstrated to be a useful route to administer pentobarbital [30]) avoids these disadvantages [31,32]. Witepsol^®^ W25 was chosen as the base for these suppositories because of its melting point (between 32 and 35.5 °C) and its high hydroxyl values (between 20 and 50 mg KOH/g) [33], which theoretically could have helped to solubilize pentobarbital. Unfortunately, this was not the case experimentally. In formulation F2, oleic acid was used to improve the pentobarbital solubility in Witepsol^®^ W25. This amphiphilic chemical agent was chosen because of its physicochemical properties: as a fatty acid (which means it is soluble in a lipophilic base), the presence of an acid function allowed a clear solution to be obtained. This could be explained by the transformation of sodium pentobarbital to the molecular state by protonation. Indeed, because oleic acid has a carboxylic acid function (pKa = 5.02), pentobarbital initially in the anionic form present in Witepsol^®^ W25 captures the proton of the carboxylic acid function and thus changes into its molecular form, thus increasing its lipophilicity [34,35]. Furthermore, its presence as a fatty acid in our diet is in favour of its security for paediatric patients [36]. Its use as an excipient in pharmaceuticals and as an emulsifying or solubilizing agent in aerosol products has also been commonly reported [37]. We demonstrated that the use of oleic acid enabled an improvement in the ease of preparation of pentobarbital suppositories, as we found 100% compliance with the 2.9.40 monography of the Ph. Eur. by four operators with no experience whatsoever in preparing suppositories, versus 25% when F1 was prepared by the same operators. However, the preparation of pentobarbital suppositories without oleic acid remains possible but needs increased technical practice.

In this study, we found a difference in the physical properties between the formulations of pentobarbital suppositories. In fact, formulation F2 containing oleic acid possessed a lower softening time (less than 3.5 min for F2 vs. about 10 min for F1), lower resistance to rupture (1 kg for F2 vs. 4.5 kg for F1), and an accelerated disintegration time (2.5 to 3 min for F2 vs. 6 to 7 min for F1). All these values are comparable to those of suppositories containing other drugs, such as acetaminophen [38], sulpiride [33], or indomethacin [39]. They are also in favour of a rapid dissolution and thus a rapid action [40] of F2 compared to F1. Sedatives are used in children whenever an examination requires the patient’s immobility. This is particularly the case for image acquisition during magnetic resonance imaging or auditory evoked potentials, for example. The quality of the examination is related to a rapid onset, a moderate duration of action, and a minimisation of adverse effects [3]. For this reason, rectal solutions and hydrogels were developed to shortcut the lag time due to suppository dissolution [16], but suppositories have the advantage of being easy to handle and do not need a device to be administered. Although other formulations containing pentobarbital have been described in the literature, all of them contain at least one ingredient that is not recommended for paediatric patients. For example, the US2538127A patent formula [41] contains propylene glycol monostearate, for which safety of use with infant and children is still debatable [42]. Despite the need for such a product, rectal dosage forms are currently not marketed, and pharmaceutical compounding remains the only and indispensable way to treat patients with pentobarbital using the rectal route. In this work, we used only safe and easily available components to prepare the pentobarbital suppositories. The development of a new pentobarbital sodium suppository drug dosage form is critical although batch sizes are limited (benchtop batch), and the quality must be well defined and evaluated. In addition, the scale up of this formulation could be performed in the case of shared production between different hospitals. Lastly, we demonstrated that we reached a total dissolution of pentobarbital in its base using oleic acid in the F2 formula, which could improve the manufacturability and could enable industrial production if desired. This work therefore presents an easy and theoretically safe solution for the preparation of pentobarbital suppositories for paediatric use.

To conduct the stability study, an existing analytical method for pentobarbital identification and quantification [25] was used to assess the physicochemical stability of our pentobarbital suppositories, after being adapted. The choice of the quantification wavelength at 214 nm assured maximum sensitivity with minor influence of the mobile phase and excipients, and the 1/10th dilution of the extracted aqueous phase before chromatographic analysis also favoured the detection of minute quantities of breakdown products in the pentobarbital suppositories. Pentobarbital solutions were analysed after forced degradation conditions, and BPs were then monitored. The detected BPs had different retention times compared to the ones found in previous studies, such as Ajemni et al. [25], which is normal as the liquid chromatography method was modified. The forced degradation results are consistent with previous studies that demonstrated a strong pentobarbital resistance to multiple stress conditions [24,25,43,44]. Nevertheless, some variations were noted in terms of the results found in forced degradation studies. Indeed, Ajemni et al. found a 49.1% pentobarbital degradation after exposition to oxidative stress (3% H_2_O_2_, 50 °C, 48 h), whereas with the same experimental conditions, we found no signs of pentobarbital degradation, even after 7 days of contact. A stronger stress condition was therefore investigated (with H_2_O_2_ 30%, 50 °C for 7 days), but it too did not induce any pentobarbital degradation (Table 4). This reproducibility limitation could be attributed to variations in protocol (sample preparation) but could be investigated further. Other stress conditions were consistent with our findings.

During the stability study, the quantity of pentobarbital in the 30 mg suppositories stayed within the 90–110% range of the initial pentobarbital quantity, for both formulations. For the 60 mg formulation, it is noteworthy that an important variation of the pentobarbital concentration (in % of T0 quantity) was observed with formulation F1, as seen in Figure 4. Indeed, at several times during the study, the mean quantity of pentobarbital quantified in the tested suppositories was above the 90–110% range of the initial pentobarbital quantity and could therefore be interpreted as a sign of instability. However, no breakdown products were detected. This could be explained by an inhomogeneity of the initial sampling of the ten units of 60 mg pentobarbital suppositories without oleic acid, especially as the mean quantity in the units analysed at the start of the study was lower than the theoretical value (58.29 mg versus 60 mg). The EAV calculated at day 1 for our batches produced supports this hypothesis of inhomogeneity because it was the highest for this 60 mg pentobarbital series without oleic acid (0.1052). For both formulations and dosage forms, after 41 weeks of storage, all pentobarbital peaks remained pure, as demonstrated by the peak purity analysis of the HPLC data (see Appendix A). The shelf life was evaluated in refrigerated conditions (5 ± 3 °C) to preserve the suppositories’ hardness as the excipients, especially oleic acid, are sensitive to temperature and tend to soften when the temperature increases. Nonetheless, regarding the chromatogram analysis, we observed new peaks for the F2 suppositories, as seen in Figure 5. This could possibly be due to a degradation of the oleic acid because these new peaks possessed retention times that were different to those of the BPs detected during the forced degradation tests and their UV spectra did not present any similarities to that of pentobarbital. Further studies investigating the stability of oleic acid would be necessary to assess and characterize its degradation over time. Without this confirmation, the shelf life of formulation F2 should be limited to 28 weeks (approximatively 7 months). On the other hand, this study demonstrated that formulation F1 did not present any signs of pentobarbital degradation for 41 weeks (approximatively 10 months).

Overall, this stability study presents strong data in favour of the physicochemical stability of both formulations of 30 and 60 mg pentobarbital suppositories when conditioned in plastic moulds and stored at refrigerated temperatures. The stability of pentobarbital in aqueous solutions has been studied, mainly for liquid injectable and rectal usage [24,44], but there were no relevant data regarding the stability of solid, suppository forms. Our study allows us to address this data weakness. The most recent rectal form study published was about a rectal hydrogel. The authors reported less than a 5% loss of pentobarbital after 90 days at 2 to 8 °C and 22 to 25 °C and did not mention the detection of any breakdown product [16].

However, clinical investigations of these developed formulae need to be conducted in order to confirm their adequacy for PPS for paediatric patients. As oleic acid is reported to be a penetration enhancer [45], it could amplify and accelerate the sedation caused by pentobarbital. For this reason, a clinical study should be conducted with a pharmacokinetic characterisation in order to confirm the safety and the efficiency of these suppositories.

## 5. Conclusions

In this work, we studied two formulations of pentobarbital sodium suppositories for paediatric procedural sedation. The formulation containing oleic acid presented a rapid disintegration, which could be in favour of rapid sedation. We demonstrated a stability of pentobarbital sodium suppositories for at least 41 and 28 weeks for the formulations without and with oleic acid in Witepsol^®^ W25, respectively, when stored at 5 ± 3 °C. A clinical study would be necessary to complete the in vivo impact of the new formulation of pentobarbital suppositories and to verify their safety.

## Figures and Tables

**Figure 1 pharmaceutics-15-00755-f001:**
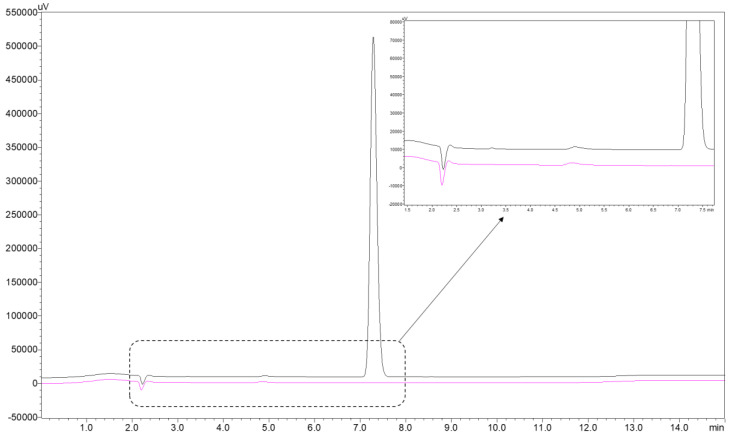
Reference chromatogram of pentobarbital sodium at a concentration of 150 µg/mL (black curve) and blank sample chromatogram (pink curve).

**Figure 2 pharmaceutics-15-00755-f002:**
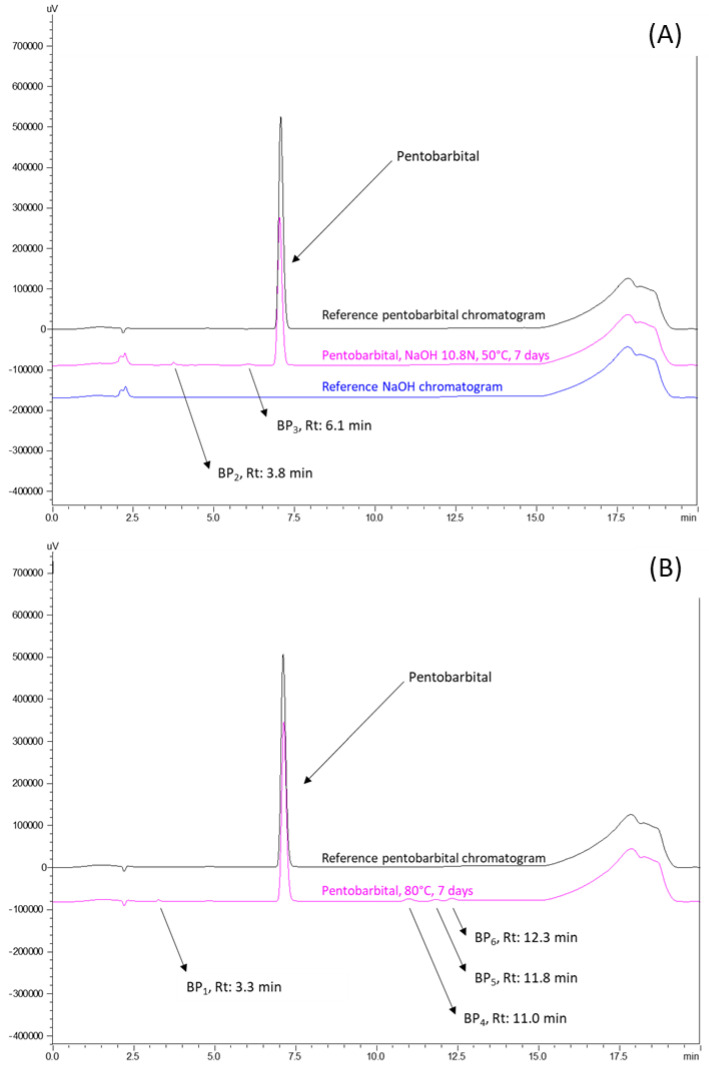
Chromatograms at 214 nm of breakdown products (BPs) obtained after forced degradations: (**A**) alkaline conditions of NaOH 10.8N for 7 days at 50 °C and (**B**) thermic conditions of 80 °C for 7 days. Rt: retention time.

**Figure 3 pharmaceutics-15-00755-f003:**
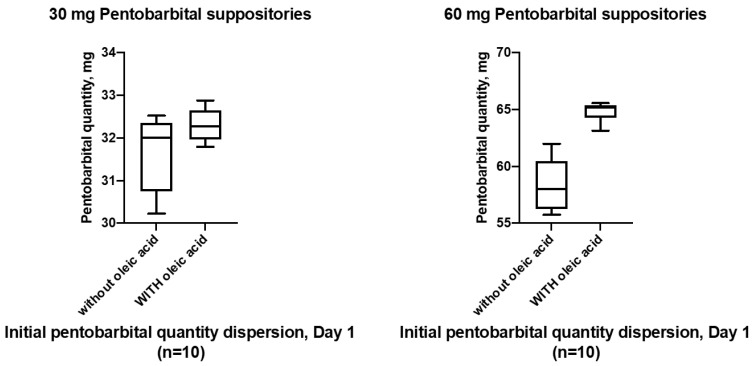
Initial distribution of pentobarbital quantity in the ten 30 mg and 60 mg pentobarbital suppositories analysed at day 1 (initial reference time point for the stability study).

**Figure 4 pharmaceutics-15-00755-f004:**
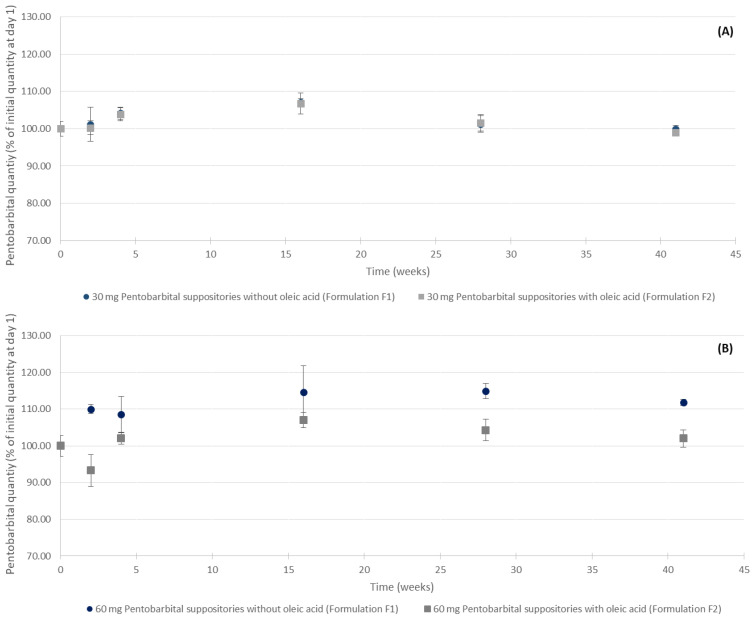
Evolution of the pentobarbital quantities of (**A**) 30 mg and (**B**) 60 mg suppositories with and without oleic acid. *n* = 5, mean ± 95% confidence interval.

**Figure 5 pharmaceutics-15-00755-f005:**
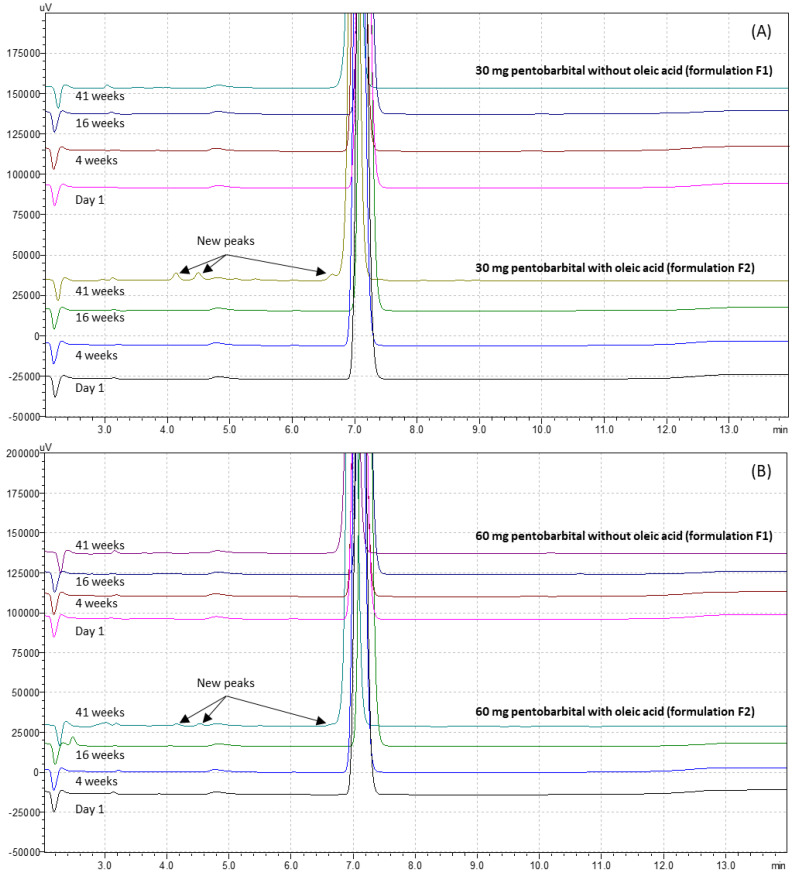
Example of pentobarbital chromatograms after storage, for (**A**) 30 mg pentobarbital suppositories and (**B**) 60 mg pentobarbital suppositories, for both studied formulations.

**Table 1 pharmaceutics-15-00755-t001:** Composition of pentobarbital suppositories (quantities for one suppository).

Components	Formulation 1				Formulation 2			
Pentobarbital sodium *(corresponding to pentobarbital base)*	30 mg *27 mg*	40 mg *36 mg*	50 mg *46 mg*	60 mg *55 mg*	30 mg *27 mg*	40 mg *36 mg*	50 mg *46 mg*	60 mg *55 mg*
Witepsol^®^ W25	1.070 g	1.060 g	1.050 g	1.040 g	0.861 g	0.830 g	0.801 g	0.775 g
Oleic acid	-	-	-	-	0.223 g	0.253 g	0.280 g	0.304 g

**Table 2 pharmaceutics-15-00755-t002:** Gradient used for the liquid chromatography mobile phase.

Time (min)	Mobile Phase (%)	
	A	B
0	40	60
12.5	40	60
15	5	95
16	5	95
16.5	40	60
20	40	60

**Table 3 pharmaceutics-15-00755-t003:** Initial characterisation of formulations F1 and F2. Values expressed as the mean ± SD (standard deviation), with n = 3, except if stated otherwise); EAV = experimental acceptance value.

Dosage	Formulation 1					Formulation 2				
	Uniformity of Dosage Units		Softening Time (s) ± SD	Resistance to Rupture (kg)	Disintegration Time (s)	Uniformity of Dosage Units		Softening Time (s)	Resistance to Rupture (kg)	Disintegration Time (s)
	Measured Content (mg), *n* = 30	Average EAV of 3 Batches (%)				Meseared Content (mg), *n* = 30	Average EAV of 3 Batches (%)			
30 mg	28.5 ± 1.1	11.1 ± 3.5	495 ± 25	3.3 ± 0.3	379 ± 9	30.9 ± 1.16	9.9 ± 2.2	203 ± 15	0.9 ± 0.1	154 ± 5
40 mg	39.4 ± 2.0	9.5 ± 4.1	550 ± 20	4.3 ± 0.1	395 ± 42	41.2 ± 1.24	9.4 ± 1.0	197 ± 15	1.0 ± 0.00	154 ± 5
50 mg	48.0 ± 2.0	10.5 ± 4.6	487 ± 35	4.5 ± 0.1	386 ± 2	50.5 ± 1.91	5.5 ± 3.1	170 ± 10	1.0 ± 0.00	176 ± 5
60 mg	61.2 ± 2.5	8.7 ± 3.4	483 ± 6	3.4 ± 0.2	399 ± 5	58.9 ± 1.33	6.2 ± 4.4	167 ± 6	0.9 ± 0.1	171 ± 5

**Table 4 pharmaceutics-15-00755-t004:** Uniformity of dosage results from the suppositories prepared by inexperienced operators. EAV: experimental acceptance value (target: <15%). n = 10, values expressed as mean ± standard deviation.

	Formulation 1		Formulation 2	
	Measured Content (mg), *n* = 10	EAV (%)	Measured Content (mg), *n* = 10	EAV (%)
Operator 1	24.93 ± 1.99	60.1%	31.32 ± 0.69	7.7%
Operator 2	25.79 ± 0.529	21%	31.16 ± 0.41	4.4%
Operator 3	28.45 ± 0.32	8.8%	31.4 ± 0.31	5.7%
Operator 4	27.19 ± 1.03	24.4%	31.35 ± 0.5	7%

**Table 5 pharmaceutics-15-00755-t005:** Forced degradation results. BP: breakdown product.

Forced Degradation Conditions	Pentobarbital Degradation (%)	Breakdown Product Retention Times
Acid: HCl 10.8 N, 50 °C, 8 days	0%	-
Alkaline: NaOH 10.8 N, 50 °C, 48 h	9.7%	BP_2_: 3.8 min; BP_3_: 6.1 min
Alkaline: NaOH 10.8 N, 50 °C, 7 days	30.9%	BP_2_: 3.8 min; BP_3_: 6.1 min
UV Light, 5.83 Mlux, 25 °C, 7 days	0%	-
Oxidative H_2_O_2_ 30%, 50 °C, 7 days	0%	-
Thermic 80 °C, 7 days	11.2%	BP_1_: 3.3 min; BP_4_: 11.0 min; BP_5_: 11.8 min; BP_6_: 12.3 min

## Data Availability

Full raw data have been provided in the Appendix A.

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
