# Peer review of "Development and Stability of a New Formulation of Pentobarbital Suppositories for Paediatric Procedural Sedation"

_pharmaceutics, 2023, doi:10.3390/pharmaceutics15030755_

Round 1

Reviewer 1 Report

Dear Sirs,

Comments and suggestions for improving the paper are included in the text.

Reviewer 2 Report

The current research on developing pentobarbital sodium-based suppositories for pediatric application is interesting. However, many areas of the manuscript need to be improved for proper flow of content and better understanding. The current version of the manuscript cannot be accepted for publication. 

  1. The manuscript needs extensive English editing.
  2. The quality and content of methodologies need to be improved for better understanding to the readers.  
  3. Please rewrite the content of the abstract. It needs to be better represented, and the also authors mentioned that formulation F1 was harder to prepare and the disintegration time was high compared with F2. At the same time, the authors also stated that several new peaks were observed for the F2 formulation. Please make it clear which formulation is stable. 
  4.  Line 24: correct uniformity content to content uniformity. 
  5. Line 25: Please avoid statements such as "harder to prepare."
  6. Line 101: what do authors mean by "realise 1.1g suppositories." Please modify as required. Similarly, please modify as required in line 104, "realised by calculating"
  7. Table 1: How much pentobarbital is present in 30 mg of pentobarbital sodium? Did authors measure the drug content for pentobarbital or pentobarbital sodium? How much was weighed for preparing formulations?
  8. Maintain uniformity for pentobarbital sodium throughout the manuscript. 
  9. Section 2.3.4: How was drug content estimated? Please elaborate. 
  10. The conclusion needs to be improved for better understanding. 

Reviewer 3 Report

The present paper is aimed at describing two alternative formulations for pentobarbital suppositories in terms of ease of preparation, content uniformity, and stability and compares the properties to some other dosage form.

While the paper describes the performed preparation procedures well and the results are evaluated correctly, I cannot see sufficiently ambitious scientific goal to justify a scientific paper. The authors themselves admit pentobarbital suppositories are used in therapy although they are not marketed, and they seem to claim the lack of marketed product being a reason for publishing their results. In my opinion such claim is not justified as the reason for no pentobarbital suppositories being marketed is more likely the low demand for such product than any challenge in their formulation. One can find a patent document US2538127A dating back to 1948 describing a suppository formulation of pentobarbital and the hospital pharmacies are able to formulate them routinely to my knowledge. Therefore, the paper does not disclose any critical formulation improvements that would be beyond the common practice in preparing personalized dosage forms.

From the different point of view, the paper scope is limited to laboratory preparation only, so that it cannot be considered as providing some industrially important results related to manufacturability of the suppository formulation.

Lastly, the paper is very specific to pentobarbital formulation and does not offer any generalization to make it interesting to broader scientific community. Honestly, comparing basic pharmacopeial quality of two formulations only with/without oleic acid addition may be hardly enough for a diploma thesis, but certainly not enough to present for international scientific audience.

Therefore, I consider the paper unsuitable for publication in Pharmaceutics

Round 2

Reviewer 2 Report

All the comments are well addressed with proper justification and supporting literature. The revised version of the manuscript can be accepted for publication. 

Author Response

The authors thank the reviewer for his comments.